# Acute renal failure in children. Multicenter prospective cohort study in medium-complexity intensive care units from the Colombian southeast

Jaime M. Restrepo[1], Mónica V. Mondragon [2], Jessica M. Forero-Delgadillo [1]*, Rubén E. Lasso[4], Eliana Zemanate[2‡], Yessica Bravo[2‡], Gastón E. Castillo[3‡], Stefany Tetay[3‡], Natalia Cabal[2‡], José A. Calvache[5,6]

1 Department of Pediatric Nephrology, Fundación Valle del Lili, Cali, Colombia, 2 Department of Pediatrics, Universidad del Cauca, Popayán, Cauca, Colombia, 3 Hospital Infantil Club Noel de Cali, Cali, Valle del Cauca, Colombia, 4 Clínica la Estancia, Popayán, Cauca, Colombia, 5 Department of Anesthesiology, Universidad del Cauca, Popayan, Cauca, Colombia, 6 Department of Anesthesiology, Erasmus University Medical Centre Rotterdam, Rotterdam, The Netherlands

☯ These authors contributed equally to this work.
‡ These authors also contributed equally to this work.
* jessicaforero@unicauca.edu.com

**Data Availability Statement:** All relevant data are within the manuscript and its Supporting Information files.

## Abstract

### Background

Acute kidney injury is frequent in critically ill children; however, it varies in causality and epidemiology according to the level of patient care complexity. A multicenter prospective cohort study was conducted in four medium-complexity pediatric intensive care units from the Colombian southeast aimed to estimate the clinical prognosis of patients with diagnosis of acute kidney injury.

### Methods

We included children >28 days and <18 years of age, who were admitted with diagnosis of acute kidney injury classified by Kidney Disease Improving Global Outcomes (KDIGO), during the period from January to December 2017. Severe acute kidney injury was defined as stage 2 and stage 3 classifications. Maximum KDIGO was evaluated during the hospital stay and follow up. Length of hospital stay, use of mechanical ventilation and vasoactive drugs, use of renal replacement therapy, and mortality were assessed until discharge.

### Results

Prevalence at admission of acute kidney injury was 5.2% (95%CI 4.3% to 6.2%). It was found that 71% of the patients had their maximum KDIGO on day one; an increment in the maximum stage of acute kidney injury increased the pediatric intensive care unit stay. Patients with maximum KDIGO 3 were associated with greater use of mechanical ventilation (47%), compared with maximum KDIGO 2 (37%) and maximum KDIGO 1 (16%). Eight

**Funding:** The authors declare that this manuscript received financial support from the ISN research center – Sister Renal Center Program-TRIO (Boston Children´s Hospital, Fundación Valle del Lili and Hospital Susana Lopez de Valencia). In total we receive 3000 us for transportation and stationery expenses.

**Competing interests:** The authors have declared that no competing interests exist.

patients with maximum KDIGO 2 and 14 with maximum KDIGO 3 required renal replacement therapy. Mortality was at 11.8% (95%CI 6.4% to 19.4%).

## Conclusion

Acute kidney injury, established and classified according to KDIGO as severe and its maximum stage, was associated with worse clinical outcomes; early therapeutic efforts should focus on preventing the progression to severe stages.

## Introduction

Acute kidney injury (AKI) is a syndrome defined by a rapid increase in serum creatinine, decrease in urine output, or both [1]. It is currently classified by following the Kidney Disease Improving Global Outcomes (KDIGO classification) [2–5]. It occurs frequently in hospitalized pediatric patients and in greater numbers in critically ill patients. The disease is associated with morbidity, prolonged hospital stay, and high risk of mortality [6–8].

Incidence and prevalence of AKI in children are widely variable due to multiple factors that influence upon the development and the course of the disease. The AWARE study reported 26% (95% CI, 25.6 to 28.2) incidence of global AKI and 11% (95% CI, 10.7 to 12.5) of severe AKI developed during the first week of hospital stay in pediatric intensive care [9]. In Colombia, studies have described an overall incidence from 5% to 11.5% and associated mortality from 31.8% to 53% [10–13].

Potential causes of AKI in pediatric intensive care units (PICU) vary according to demographic or health characteristics and differ according to the level of complexity of care. Therefore, in highly complex PICU, the main characteristics described are sepsis, organ transplants, and cardiovascular surgery [10,11,13]. Sepsis is one the main causes of AKI in children and its incidence has been described between 9% and 34% [14–17]. Additionally, in medium- and low-complexity levels, hypotension, dehydration, and sepsis are the most important factors [12,18,19]. In Colombia, retrospective studies have been carried out in high-complexity PICU [10–13] seeking to stablish the outcomes of AKI patients. Prognosis has been related with age of the patients, presence of infection, use of inotropic agents and mechanical ventilation, as well as, nutritional and socioeconomic status, and availability of health care services [10–13]. However, lack of unified data is still the case regarding epidemiology distribution and outcomes in Colombia and information on pediatric population is available mainly from high-complexity hospitals in big cities [20].

The AKI Committee of the Latin American Society of Nephrology and Hypertension (SLANH, for the term in Spanish) conducted an epidemiological study in Latin America and the Caribbean to assess the AKI profile according to geographic distribution and its relationship with social conditions. It included 905 patients, but only reports data from 21 patients who were under 18 years of age and almost no patients from Colombia. By using the KDIGO classification, the authors reported factors, such as dehydration, presence of shock, and use of nephrotoxic drugs as the most common potential causes of AKI. Mortality rate has been related with shock status, major cardiac abnormalities, presence of sepsis, use of renal replacement therapy (RRT), and use of mechanical ventilation [21].

The aim of this study was to describe the prognosis of AKI patients admitted to PICU, besides associated complications and mortality in medium-complexity levels of care (II and III) from the Colombian southeast. This study was conducted by adhering to the strategy

proposed by the International Society of Nephrology (ISN): zero preventable deaths by 2025 (0by25) and aimed mainly at developing countries [19].

## Methods

### Study population and settings

A multi-center prospective cohort study was conducted between January and December of 2017, including all children >28 days and <18 years of age who were admitted to four PICU (46 intensive care levels II-III beds in total) from the Colombian southeast (Club Noel Hospital in Cali, San José University Hospital, Susana Lopez de Valencia Hospital, and La Estancia Clinic in Popayán). After a comprehensive initial evaluation, we included patients with AKI diagnosis through KDIGO classification and admitted to PICU during this period. Patients with known structural or congenital renal abnormality, chronic kidney disease, and history of renal transplant were excluded.

The study was conducted in accordance with the Declaration of Helsinki and the study protocol was approved by the ethics committees in each institution: San José University Hospital (approval number R-27-10-16); Susana Lopez de Valencia Hospital (approval number R- 20-06-17); La Estancia Clinic (approval number R-30-05-17); and Club Noel Hospital (approval number R-24-10-17). According to Colombian regulations, the study was classified in the category of "observational research without risk" because there was no intervention or intentional modification of the biological, physiological, psychological, or social variables of the participants. Therefore, the ethics committees waived the informed consent requirement.

### Definitions and measurements

Acute kidney injury was defined with the KDIGO renal damage classification, which considers diuresis over time (hours) and assesses increased creatinine values with respect to the patient's baseline value. To determine AKI, serum creatinine was measured on admission to hospital and PICU and daily until discharge and/or death. Continuous monitoring was performed of diuresis, classifying patients with the lowest value of diuresis and the highest rise of the value of serum creatinine reached during their whole hospital stay. The maximum KDIGO was registered in a physical form and, subsequently, with an electronic database, which was registered weekly by the physician assigned by each participating institution. Finally, a revision was carried out of all the data, which allowed to fill in anything missing, debug, and analysis.

Acute kidney injury stage 1 corresponds to increased serum creatinine >0.3 mg/dl or elevation of 1.5–1.9 times with respect to the patient's baseline or diuresis <0.5 ml/kg/h for 6 to 12 h); stage 2 (elevation of 2–2.9 times the creatinine baseline value or diuresis <0.5 ml/kg/h for >12 h), and stage 3 (elevation of baseline creatinine ≥3 times, acute increase >4 mg/dl or use of RRT or diuresis <0.3 ml/kg/h for >24 h or anury >12 h). Severe acute kidney injury corresponded to patients classified as KDIGO 2 and 3. Baseline creatinine was the lowest value existing during six months prior to admission to PICU or that available at the emergency department.

Patients were initially screened by medical history, laboratory results and inclusion and exclusion criteria by pediatric intensivists at each PICU. Infant demographic data were collected, including age, gender, ethnic group, rural versus urban origin, social security, nutritional status, on admission and primary diagnosis group at PICU admission. Clinical characteristics and laboratory data were documented at admission and daily until discharge or death.

The main outcome under study was mortality at PICU stay. Secondary outcomes were the appearance of maximum AKI stage observed during the first seven days after PICU admission, length of PICU stay, use and duration of mechanical ventilation, use of vasoactive medications and diuretics, and requirement of RRT.

## Statistical analysis

Descriptive statistics were used to analyze the demographic data and the distribution of each variable within the sample. Values are presented as mean and standard deviations or median with interquartile range. Categorical variables were described by using absolute frequencies and percentages.

This study was planned to estimate mortality of patients admitted to PICU with diagnosis of AKI. Published national data have estimated mortality ranging widely from 43% to 50% [10–13] with substantial variability among studies due to populations considered. To measure this proportion within a 10% margin of error and an expected 40% mortality, we estimated a sample size of 93 patients without dropout rate.

We first calculated the global prevalence and confidence interval (95%CI) of AKI in critically ill children during the study period. Patients with AKI diagnosis were classified into KDIGO categories (stages 1, 2, and 3) and stage 2 and 3 as severe AKI. In addition, maximum KDIGO stage during the total PICU stay was quantified and presented as frequencies and proportions. KDIGO at admission and maximum KDIGO stage during PICU stay were used to analyze mortality and other related outcomes.

Cumulative incidence of outcome variables of RRT, use of vasoactive drugs, use of diuretics, use of mechanical ventilation, days of mechanical ventilation, mortality, and length of stay at PICU were presented and stratified by type of AKI upon admission (AKI and severe AKI). Categorical outcomes were compared by using Chi-squared test and time-to-event outcomes using log rank test.

To analyze time-to-event data, we used survival analysis and the Kaplan-Meier method. Statistical significance was determined by using the 95% CI and a $P$ value of 0.05. All analyses were performed by using STATA software (StataCorp, 2011, Stata Statistical Software: Release 12. College Station, TX, USA) and R [22,23].

## Results

We screened 2,120 patients admitted to four PICU during the recruitment period (HSLV n = 631, HUSJ n = 353, Club Noel n = 611, Clínica La Estancia n = 525). Patient demographic data and clinical characteristics are shown in Table 1.

According to the KDIGO definition, AKI was diagnosed in 110 patients with 5.2% (95% CI 4.3% - 6.2%) prevalence. Of the patients diagnosed, 52 (47%) had stage 1 AKI, 34 (31%) had stage 2 AKI, and 24 (22%) had stage 3 AKI. Severe AKI (KDIGO stage 2 and 3) was developed in 58 patients (2.7%; 95%CI 2.1–3.5) and 78 patients (71%) reached their maximum KDIGO stage on the first day of admission to PICU.

Of the 110 patients with AKI, 22 (20%) received RRT during follow-up. Hemodialysis was the most-used modality of RRT in 13 patients followed by peritoneal dialysis in nine patients. No significant association was noted for gender, age, primary diagnosis, and nutritional status with the presence of severe AKI (Table 2).

## Association between severity of acute kidney injury on admission to PICU and mortality

The mortality rate for patients with severe AKI was higher than in mild/moderate cases. There were five deaths in patients with stage 1 AKI at admission, three deaths in stage 2 AKI, and five deaths in stage 3 AKI (Table 2). Death occurred in eight of the 58 patients with severe AKI, compared with five of the 52 patients without severe AKI ($P$ <0.498). Few events prevent further analysis of this outcome.

**Table 1. Demographic data and clinical characteristics of patients.**

| Variables | Value |
|---|---|
| Male sex | 63 (57.3%) |
| Age on admission (years) | Mean 4.1 |
| | SD 4.6<br>Median 2 |
| | IQR 0.56 to 7.0 |
| Age younger than one year | 62 (56.4%) |
| Race or ethnic group | |
| Mestizo | 59 (53.6%) |
| Afro-Colombian | 20 (18.2%) |
| Indigenous | 20 (18.2%) |
| Other | 11 (10%) |
| Place of residence | |
| Rural | 50 (45.5%) |
| Urban area | 60 (54.5%) |
| Social security | |
| Contributive | 20 (18.2%) |
| Subsidized | 85 (77.3%) |
| Other | 5 (4.5%) |
| Nutritional status on admission | |
| Malnutrition | 27 (24.5%) |
| At risk of malnutrition | 16 (14.5%) |
| Primary diagnosis group at ICU admission | |
| Infectious | 40 (36.4%) |
| Respiratory | 34 (30.9%) |
| Renal | 9 (8.2%) |
| Surgical or trauma | 9 (8.2%) |
| Cardiovascular | 5 (4.5%) |
| Central nervous system | 4 (3.6%) |
| Burns | 4 (3.6%) |
| Metabolic | 2 (1.8%) |
| Immunologic | 2 (1.8%) |
| Poisoning | 1 (0.9%) |
| PRISM classification | |
| Low | 39 (35.5%) |
| Moderate | 28 (25.5%) |
| High | 15 (13.6%) |
| Very high | 28 (25.5%) |
| Potential physiopathological mechanism of AKI | |
| Pre-renal AKI | 90 (81.8%) |
| Renal AKI | 20 (18.2%) |

## Maximum stage of acute kidney injury during PICU stay and outcomes

An increase in the severity stage of AKI showed an incremental length of PICU stay ($P$ <0.001) (Fig 1). Median length of PICU stay was nine days for patients with maximum KDIGO stage 1 AKI; 10 days for patients with maximum KDIGO stage 2 AKI, and 12 days for patients with maximum KDIGO stage 3 AKI during PICU stay.

**Table 2. Outcomes in patients with AKI diagnosis stratified by KDIGO stage, mild/moderate AKI and severe AKI on admission.**

| | Total cohort (n = 110) | AKI KDIGO stage 1 Mild/moderate AKI (n = 52) | AKI KDIGO stage 2 (n = 34) | AKI KDIGO stage 3 (n = 24) | P-value | AKI KDIGO stages 2 and 3. Severe AKI (n = 58) | P-value[a] |
|---|---|---|---|---|---|---|---|
| Renal-replacement therapy | 22 (20%) | 10 (19.2%) | 6 (17.6%) | 6 (25%) | 0.774 | 12 (20.7%) | 0.849 |
| Use of vasoactive drugs | 78 (71%) | 38 (73.1%) | 23 (67.6%) | 17 (70.8%) | 0.863 | 40 (69%) | 0.635 |
| Use of diuretics | 88 (80%) | 43 (82.7%) | 28 (82.4%) | 17 (70.8%) | 0.446 | 45 (77.6%) | 0.504 |
| Use of mechanical ventilation | 76 (69%) | 38 (73.1%) | 22 (64.7%) | 16 (66.7%) | 0.684 | 38 (65.5%) | 0.392 |
| Days of mechanical ventilation[b] | 5 (3.4–6.5) | 7 (4.3–9.6) | 4 (1.7–6.2) | 4 (1.7–6.2) | 0.322[c] | 4 (2.3–5.6) | 0.183[c] |
| Length of stay at PICU[b] | 12 (9.2-14-7) | 14 (12.5–15.4) | 7 (3.2–10.7) | 11 (8.6–13.3) | 0.092[c] | 10 (6.8–13.1) | 0.343[c] |
| Mortality[d] | 13 (11.8%) | 5 (9.6%) | 3 (8.8%) | 5 (20.8%) | 0.342 | 8 (13.8%) | 0.498 |

[a]Comparison between mild/moderate AKI versus severe AKI.

[b]Median and [IQR].

[c]p-value from log rank test.

[d]Overall mortality 11.8% 95%CI 6.4% - 19.4%.

Patients with a maximum KDIGO stage 3 AKI during PICU stay was associated with higher use and duration of mechanical ventilation in comparison with patients with maximum stages 1 and 2. Use of mechanical ventilation in patients with a maximum stage 3 AKI was 47% compared with maximum stage 2 with 37% and maximum stage 1 with 16% ($P < 0.021$). The mean time of mechanical ventilation was 4.7 days for patients with a maximum KDIGO stage 1 AKI, 5.7 days for a maximum KDIGO stage 2, and 8.5 days for a maximum KDIGO stage 3.

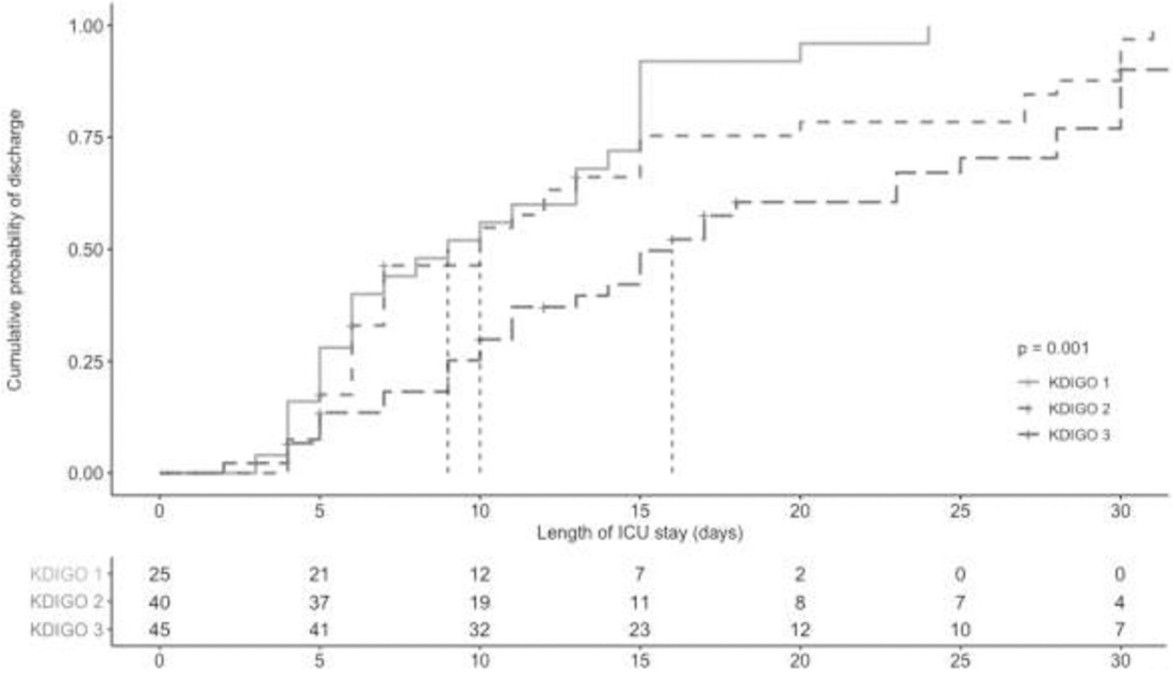

**Fig 1. Median length of PICU stay stratified by maximum KDIGO stage during PICU stay.**

An increase in the severity of AKI stage during PICU stay was associated with increased use of RRT. No patients with maximum KDIGO stage 1 AKI required RRT in contrast with eight (36%) in maximum stage 2 and 14 (64%) in maximum stage 3 ($P <0.008$).

## Progression of acute kidney injury and center variability

The daily frequency of severe AKI was 52.7% at day 1, 41.7% at day 7, and 20% at day 14. We observed inter-center variability in the severity of AKI. At La Estancia Clinic, 18 of 20 (90%) patients were diagnosed with severe AKI; 16 of 42 (38%) at Club Noel Hospital; 16 of 31 (51.6%) at San José University Hospital; and 8 of 17 (52.7%) at Susana Lopez de Valencia Hospital.

## Discussion

This research is the first multi-center prospective cohort study conducted in Colombia in medium-complexity PICU. We found a global 5.2% prevalence of AKI at PICU, severe AKI was found in 2.7% and associated mortality of this cohort of patients was at 11.8%. Prior observational studies on AKI in Colombia have described mortality from 43% to 53% in high-complexity PICU and around 23% in medium-complexity PICU [11–13]. In a retrospective study with the new KDIGO classification, Serna-Higuita *et al.*, showed 31.8% mortality in pediatric patients with AKI in Colombia [10]. Our frequencies of AKI and mortality are lower compared to reports from developed countries [9,16,18,24], as well as Colombian reports [10]. This variability is probably due to the severity of the populations studied, level of complexity of care, and different definitions used to classify AKI patients.

Pediatric patients with AKI reach their maximum KDIGO stage during the first week after diagnosis [9,16,25,26]. In our study, most of the patients included reached their maximum KDIGO stage on day one after admission to PICU and this may be explained by several potential factors: patients included came from rural areas, had varying degrees of malnutrition, and presented insurance-related barriers to health care in agreement with previous recent reports from Latin America [21].

The area of influence of most hospitals included is the department of Cauca and this is one of the least developed and rural areas in the country and among the most affected by the armed conflict [27]. It comprises 84 indigenous areas of the 679 existing in Colombia and 23% of its population has some alteration in the nutritional status [28]. In addition, the department of Cauca has only three hospitals of moderate complexity, which generates difficulty of access to health that include long distances and long time to reach appropriate care. Considering these characteristics, our study suggests that AKI diagnosis in our population was probably late and perhaps related to disorders of volume status, infection, degrees of malnutrition, and characteristics of the Colombian health care system. Infectious diseases are the most common related causes of AKI in developing countries in children [29]. In agreement, our results and other developing countries report similar profiles upon admission [24,25].

An increase in the severity of AKI (maximum KDIGO stage during PICU stay) was associated with more use and longer duration of mechanical ventilation, as well as use of RRT in this study. A multi-national study from Asia, Australia, Europe, and North America (AWARE study), showed that an increase in the peak stage of AKI was associated with greater use of RRT and more days of mechanical ventilation [9]. Similar results have been reported by studies in medium-complexity hospitals of developing and developed countries [24,30,31]. Excluding patients undergoing cardiovascular surgery or transplant, studies have reported increased length of hospital stay: 9.7 days in AKI patients versus 4.6 days in non-AKI patients [31]. The AKI diagnosis may increase hospital stay by up to 10 days [32]. In this study, an increase in the maximum AKI stage conferred an incremental length of PICU stay.

In a recent survey carried out by SLANH in Latin America, hemodialysis was the main approach to RRT and its selection was guided mainly by economic and organizational regional reasons [21]. However, there would be strong variation among and within countries. In studies from large cities in Colombia, peritoneal dialysis has been the most-used modality of RRT [11–13], as described in other developing countries [33]. Peritoneal dialysis is widely available and it avoids the requirement of hemodialysis vascular access. Our patients received mainly hemodialysis and most were recruited in Club Noel Hospital in Cali, Colombia. This is a reference hospital for a large population and it receives critically ill patients more frequently. Administrative issues, population admitted, as well as previous experience and clinical preferences with each dialysis modality may explain these differences.

Our study has several limitations to consider when interpreting our results. First, we differentiated intrinsic renal damage and pre-renal injury according to some physiological and clinical characteristics of the underlying diseases. The most frequent etiologies were 1) infectious (which probably causes hypovolemic or renal vasoconstriction); 2) depletion of intravascular volume (due to gastrointestinal losses, such as diarrhea and vomiting); and 3) redistribution to the interstitial space (probably secondary to malnutrition). We know this strategy may potentially introduce information bias in this classification. However, it is part of our usual clinical care. Secondly, we were unable to measure the percentage of fluid overload, despite the importance of the association of fluid overload and increased morbidity and mortality in acute renal failure [34]. We lack of a systematic record of these data in the clinical setting, along with other information, *i.e.*, renal ultrasound in all patients with AKI diagnosis. Considering that, we introduced improved measures of clinical care. Third, difficulties emerged in the follow up of patients once they were discharged from the institution. We lost the possibility of contacting them for periodic assessment by pediatric nephrology and/or pediatrics and detect long-term outcomes, like the appearance of arterial hypertension or proteinuria, described widely as important markers of kidney disease. Our efforts to continue studying this population are undergoing. Finally, incidence of few events in some outcomes prevent further analysis and potential prognostic information remains under study.

In conclusion, this is the first multi-center study reporting AKI epidemiology and prognosis in pediatric patients attended at medium-complexity hospitals in a middle-income country. This cohort was recruited in hospitals regardless of major conditions, like cardiac surgeries or transplants. Severe AKI was associated to worse clinical outcomes; early therapeutic efforts should focus on preventing the progression to severe stages.

We are not adding new information about AKI at global level, but our research allows us to describe the regional and national epidemiology and to know the prognosis of AKI in pediatric Colombian patients following the aims of the 0by25 strategy from the International Nephrology Society. This strategy seeks to prevent all deaths due to AKI by 2025 and to conduct timely diagnosis and treatment of AKI to patients with reversible pathologies, especially in countries with low and very low economic resources and with limitations in access to health care [19]. We hope our contribution is in line with this strategy and improves our understanding of AKI in Latin America.

## Supporting information

**S1 Dataset.**
(XLSX)

## Acknowledgments

The authors thank Dr. Ángela Merchán, Masters in Clinical Research Applied to Health Sciences, professor at Universidad del Cauca, who participated in accompanying the technical drafting and editing of this document.

## Author Contributions

**Conceptualization:** Jaime M. Restrepo, Mónica V. Mondragon, Rubén E. Lasso, Gastón E. Castillo, Stefany Tetay, Natalia Cabal.

**Data curation:** Jessica M. Forero-Delgadillo, Rubén E. Lasso, Stefany Tetay, Natalia Cabal.

**Formal analysis:** Jaime M. Restrepo, Jessica M. Forero-Delgadillo.

**Funding acquisition:** Jaime M. Restrepo.

**Investigation:** Jaime M. Restrepo, Mónica V. Mondragon, Jessica M. Forero-Delgadillo, Rubén E. Lasso, Yessica Bravo, Gastón E. Castillo, José A. Calvache.

**Methodology:** Jaime M. Restrepo, Rubén E. Lasso, José A. Calvache.

**Project administration:** Jaime M. Restrepo, Jessica M. Forero-Delgadillo, Rubén E. Lasso, Eliana Zemanate, Yessica Bravo, José A. Calvache.

**Resources:** Jaime M. Restrepo, Rubén E. Lasso, Gastón E. Castillo.

**Software:** José A. Calvache.

**Supervision:** Jaime M. Restrepo, Jessica M. Forero-Delgadillo, Rubén E. Lasso, Eliana Zemanate, Yessica Bravo, José A. Calvache.

**Validation:** Jaime M. Restrepo, Rubén E. Lasso, José A. Calvache.

**Visualization:** Jaime M. Restrepo, Jessica M. Forero-Delgadillo, José A. Calvache.

**Writing – original draft:** Jaime M. Restrepo, Mónica V. Mondragon, Jessica M. Forero-Delgadillo, Rubén E. Lasso, Eliana Zemanate, Yessica Bravo, José A. Calvache.

**Writing – review & editing:** Jaime M. Restrepo, Mónica V. Mondragon, Jessica M. Forero-Delgadillo, Rubén E. Lasso, José A. Calvache.

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
