## [Decision Letter · Decision Letter 0]

13 Mar 2020

PONE-D-20-03806

Acute Renal Failure in Children. Multicenter Prospective Cohort Study in Medium Complexity Intensive Care Units from the Colombian Southeast

PLOS ONE

Dear Dr. Forero,

Thank you for submitting your manuscript to PLOS ONE. After careful consideration, we feel that it has merit but does not fully meet PLOS ONE’s publication criteria as it currently stands. Therefore, we invite you to submit a revised version of the manuscript that addresses all the critical points raised during the review process by the two reviewers.

We would appreciate receiving your revised manuscript by Apr 27 2020 11:59PM. To enhance the reproducibility of your results, we recommend that if applicable you deposit your laboratory protocols in protocols.io, where a protocol can be assigned its own identifier (DOI) such that it can be cited independently in the future. For instructions see: http://journals.plos.org/plosone/s/submission-guidelines#loc-laboratory-protocols

We look forward to receiving your revised manuscript.

Kind regards,

Emmanuel A Burdmann

Academic Editor

PLOS ONE

Journal Requirements:

Journal Requirements:

'This study was approved by the ethics committees at the four institutions (Resolution 301 of May 2013 and Resolution 0665 of 19 October 2011).'

(a) Please amend your current ethics statement to include the full name of the ethics committee/institutional review board(s) that approved your specific study.

(b) Once you have amended this/these statement(s) in the Methods section of the manuscript, please add the same text to the “Ethics Statement” field of the submission form (via “Edit Submission”).

3. Please provide additional details regarding participant consent. In the ethics statement in the Methods and online submission information, please ensure that you have specified (1) whether consent was informed and (2) what type you obtained (for instance, written or verbal, and if verbal, how it was documented and witnessed). If your study included minors, state whether you obtained consent from parents or guardians.

4. In your Methods section, please provide additional information about the participant recruitment method and the demographic details of your participants. Please ensure you have provided sufficient details to replicate the analyses such as: a) the recruitment date range (month and year) and b) a description of how participants were recruited.

5. Please provide a sample size and power calculation in the Methods, or discuss the reasons for not performing one before study initiation.

Reviewers' comments:

Reviewer's Responses to Questions

**Comments to the Author**

1. Is the manuscript technically sound, and do the data support the conclusions?

Reviewer #1: Yes

Reviewer #2: Yes

2. Has the statistical analysis been performed appropriately and rigorously? 

Reviewer #1: Yes

Reviewer #2: Yes

3. Have the authors made all data underlying the findings in their manuscript fully available?

Reviewer #1: Yes

Reviewer #2: Yes

4. Is the manuscript presented in an intelligible fashion and written in standard English?

Reviewer #1: No

Reviewer #2: No

5. Review Comments to the Author

Reviewer #1: This is an interesting study whose objective was to estimate the incidence and prognosis of AKI, associated complications and mortality, in medium-complexity PICUs from the Colombian southeast using KDIGO classification.

The title and abstract are clear. In the introduction the AKI definition ‘ Acute kidney injury (AKI) is a pathology defined as a sudden loss of the renal function that can be reversible.” should be changed to “Acute kidney injury (AKI) is a syndrome, defined by a rapid increase in serum creatinine decrease in urine output, or both”, (Lancet 2019; 394: 1949–64), its reversibility should not be included in the definition

Methodology and Statistical analysis are well described.

Results are very difficult to interpret:

The text states that “ stage I (mild) was diagnosed in 2.5% of total admissions (52 patients), stage II (moderate) in 1.6% (34 patients), and stage III (severe) in 1.1% (24 patients). Seventy eight patients (71%) reached their maximum KDIGO stage on the first day from the admission to PICU

Commentary: These percentages should be calculated in relation to the 110 patients diagnosed with AKI and not in relation to the total screened admissions

Table 1 shows that 90 patients (81.8% were classified as pre-renal AKI 20 (18.2%) as renal AKIl

Commentary: based on the list of primary diseases this information is probably inaccurate

In the methodology session Severe acute kidney injury corresponded to patients classified as KDIGO 2 and 3 The results session states that “Severe AKI developed in 58 patients (2.7%; 95%CI 2.1 to 3.5). information that is also on table 2 but the results description continues as follows “ The severity classification according to KDIGO was 25 patients (22.7%) in stage 1, 40 patients (36.4%) in stage 2 and 45 patients (40.9%) in stage 3, severe AKI affected in fact 40+45= 85 patients

Commentary: the authors should review their definitions and review the tables as well as the rest of the text

The phrase “On average, the severe stage of AKI was higher in non-survivors than in survivors” could be written as “The mortality rate for patients with severe AKI was higher than in the mild /moderate cases. The other parameters such as length of stay, mechanical ventilation and RRT could have their relation to AKI re-phrased in the same way English language should be revised throughout the text

The authors conclusions are in consonance with the literature but patient characteristics differ from the ones previously described in developed countries, the patients described in this study were admitted to medium complexity PICUs, mostly with medical primary diseases , 71% of the patients had their maximum KDIGO on day one of admission, so the diagnosis of AKI was probably late ( is it due to lack of hospital facilities in the Colombian rural area??) and probably related to disorders of volume status and infection , almost 40% of them were presented malnutrition or were at risk of this comorbidity, My suggestion would be for the authors to describe these characteristics in more detail and the efforts that should be developed to minimize them

Reviewer #2: The authors conducted a prospective, observational study to assess the incidence and prognosis of AKI in PICUs of southeast Colombia. They found a cumulative incidence of AKI of 5.2% and that AKI was associated with worse clinical outcomes. The study will improve the knowledge of the local epidemiology of AKI in Colombia, but will not have any impact on the knowledge of what is already know about AKI and outcomes in pediatric population.

I have some comments, and observations that the authors need to address:

1. I would recommend avoiding abbreviations on the abstract, and if used please define them (i.e. KDIGO).

2. Authors need to exclude from their analysis patients would were diagnosed with AKI at ER, and only include patient that developed AKI during PICU stay, the reasoning for this is that some % of patients with AKI diagnosis at ER could have non-diagnosed CKD, could be with AKD, or have community acquired AKI and this different type of populations have different risk for several short and long-term outcomes.

3. Why do authors exclude patients with CKD? Since CKD is an important risk factor for subsequent AKI and could affect outcomes?

4. How you deal with the fact that some patients did not had baseline serum creatinine? You used urine volume criteria? You used MDRD <60 ml/m/1.73m2 to estimate baseline sCr?

5. On table one; how you were able to differentiate patients with intrinsic renal damage from those who have Pre-renal? Please clarify. What do you refer that 18% of patients have “renal classification of AKI” this is not clear.

6. On page 8 you said that 58 patients developed severe AKI, however you later said that 45 patients developed stage 3 of AKI is that not consider severe AKI too? You need to check your numbers .

7. You need to show fluid balance of your patients, fluid overload is an important contributing factor for adverse outcomes: Fluid overload should be assessed as % increase of body weight, Bouchard et al have shown that patients with fluid overload defined as an increase in body weight of over 10% had significantly more respiratory failure, need of mechanical ventilation, and more sepsis.

Please review and use this definition for fluid overload:

Percentage of fluid overload adjusted for body weight: cumulative fluid balance that is expressed as a percent. A cutoff of ≥10% has been associated with increased mortality. Fluid overload percentage can be calculated using the following formula:

% Fluid overload = ((total fluid in – total fluid out)/admission body weight x 100)

Authors should stratified each group of patients in < 10% of fluid overload, 11% to 20% of fluid overload, > 20% of fluid overload for example, in order to assess for adverse outcomes.

Review this study: Bouchard J, Mehta RL. Fluid balance issues in the critically ill patient. Contrib Nephrol. 2010;164:69-78.

8. What was the mean or median of fluid received during PICU stay? Would like to see fluids that were given by oral route and by I.V.

9. I recommend also that authors need to look at factors that could have influence on having prolonged mechanical ventilation like sepsis, comorbidities (i.e. chronic illness), nutrition and metabolic problems, cardiovascular disease, pneumonia, and ventilation modality.

6. PLOS authors have the option to publish the peer review history of their article (what does this mean?). If published, this will include your full peer review and any attached files.

Reviewer #1: Yes: Vera H. Koch

Reviewer #2: No

---

## [Author Response · Author response to Decision Letter 0]

11 May 2020

General comments

1. Please ensure that your manuscript meets PLOS ONE's style requirements, including those for file naming. The PLOS ONE style templates can be found at:

http://www.plosone.org/attachments/PLOSOne_formatting_sample_main_body.pdf and http://www.plosone.org/attachments/PLOSOne_formatting_sample_title_authors_affiliations.pdf

R/ Done. Revised in detail. 

We have followed PLOS ONE's style requirements in all sections (Abstract, introduction, materials and methods, results, conclusion, acknowledgments, references and supporting information). 

Author names and affiliations and all used symbols were checked and corrected.

R/ Done. We added all information regarding specific dates of ethical approval at each site and expanded the corresponding paragraph. 

Quote: “The study was conducted in accordance with the Declaration of Helsinki and the study protocol was approved by the ethics committees in each institution: San José University Hospital (approval number R-27-10-16); Susana Lopez de Valencia Hospital (approval number R- 20-06-17); La Estancia Clinic (approval number R-30-05-17); and Club Noel Hospital (approval number R-24-10-17).”.

3. Please provide additional details regarding participant consent. In the ethics statement in the Methods and online submission information, please ensure that you have specified (1) whether consent was informed and (2) what type you obtained (for instance, written or verbal, and if verbal, how it was documented and witnessed). If your study included minors, state whether you obtained consent from parents or guardians.

R/ Corrected and completed. 

We provide additional details regarding ethical approval and informed consent. Final paragraph of study population and settings was modified accordingly.

Quote “The study was conducted in accordance with the Declaration of Helsinki and the study protocol was approved by the ethics committees in each institution: San José University Hospital (approval number R-27-10-16); Susana Lopez de Valencia Hospital (approval number R- 20-06-17); La Estancia Clinic (approval number R-30-05-17); and Club Noel Hospital (approval number R-24-10-17). According to Colombian regulations, the study was classified in the category of “observational research without risk” because there was no intervention or intentional modification of the biological, physiological, psychological, or social variables of the participants. Therefore, the ethics committees waived the informed consent requirement.”.

4. In your Methods section, please provide additional information about the participant recruitment method and the demographic details of your participants. 

Please ensure you have provided sufficient details to replicate the analyses such as: a) the recruitment date range (month and year) and b) a description of how participants were recruited.

R/ Done. Completed.

In the methods section we have provided additional information for the patient recruitment and demographic details. Also, we added information about specific recruitment dates.

In addition, measurement and data details are presented in the section “definitions and measurements”. 

Quote: "A multi-center prospective cohort study was conducted between January and December of 2017, including all children >28 days and <18 years of age who were admitted to four PICU (46 intensive care levels II-III beds in total) from the Colombian southeast (Club Noel Hospital in Cali, San José University Hospital, Susana Lopez de Valencia Hospital, and La Estancia Clinic in Popayán). After a comprehensive initial evaluation, we included patients with AKI diagnosis through KDIGO classification and admitted to PICU during this period. Patients with known structural or congenital renal abnormality, chronic kidney disease, and history of renal transplant were excluded.”

Additionally, we completed the section “definitions and measurements” in accordance.

Quote: “Patients were initially screened by medical history, laboratory results and inclusion and exclusion criteria by pediatric intensivists at each PICU. Infant demographic data were collected, including age, gender, ethnic group, rural versus urban origin, social security, nutritional status, on admission and primary diagnosis group at PICU admission. Clinical characteristics and laboratory data were documented at admission and daily until discharge or death.”

5. Please provide a sample size and power calculation in the Methods, or discuss the reasons for not performing one before study initiation.

R/ Completed and corrected. 

By using local and regional information regarding cumulative incidence of AKI, during our protocol stage we did several calculations of our required sample in order to estimate the mortality of patients diagnosed with AKI. Following the requirement of the reviewer, we added to the methods a detailed section a paragraph about sample size calculation.

Quote: “This study was planned to estimate mortality of patients admitted to PICU with diagnosis of AKI. Published national data have estimated mortality ranging widely from 43% to 50% [10-13] with substantial variability among studies due to populations considered. To measure this proportion within a 10% margin of error and an expected 40% mortality, we estimated a sample size of 93 patients without dropout rate.”.

Comments to the Author 

1. Is the manuscript technically sound, and do the data support the conclusions?

 Reviewer #1: yes, Reviewer #2: yes.

2. Has the statistical analysis been performed appropriately and rigorously?

Reviewer #1: yes, Reviewer #2: yes.

3. Have the authors made all data underlying the findings in their manuscript fully available?

Reviewer #1: yes, Reviewer #2: yes.

4. Is the manuscript presented in an intelligible fashion and written in standard English?

Reviewer #1: No, Reviewer #2: No.

R/ Done. 

All English language grammar and related details were revised by a professional editor to check for accuracy and intelligibility. We hope to reach the standards of the journal.

5. Review Comments to the Author

Reviewer #1

1. This is an interesting study whose objective was to estimate the prognosis of AKI, associated complications and mortality, in medium-complexity PICUs from the Colombian southeast using KDIGO classification.

The title and abstract are clear. In the introduction the AKI definition ‘Acute kidney injury (AKI) is a pathology defined as a sudden loss of the renal function that can be reversible.” should be changed to “Acute kidney injury (AKI) is a syndrome, defined by a rapid increase in serum creatinine decrease in urine output, or both”, (Lancet 2019; 394: 1949–64), its reversibility should not be included in the definition 

R/ Thank you for the comment. 

Done. 

We have adjusted the introduction following the recommendation and modified our definition to become it clear. We also added the recommended reference. 

Quote “Acute kidney injury (AKI) is a syndrome defined by a rapid increase in serum creatinine, decrease in urine output, or both”.

2. Methodology and Statistical analysis are well described.

R/ Thank you for the comment.

3. Results are very difficult to interpret:

The text states that “ stage I (mild) was diagnosed in 2.5% of total admissions (52 patients), stage II (moderate) in 1.6% (34 patients), and stage III (severe) in 1.1% (24 patients). Seventy eight patients (71%) reached their maximum KDIGO stage on the first day from the admission to PICU.

Commentary: These percentages should be calculated in relation to the 110 patients diagnosed with AKI and not in relation to the total screened admissions.

R/ Done. Results section was checked and re-organized.

We clarified our screened and included patients as well as the KDIGO stage at diagnosis.

Quote: “According to the KDIGO definition, AKI was diagnosed in 110 patients with 5.2% (95% CI 4.3% - 6.2%) prevalence. Of the patients diagnosed, 52 (47%) had stage 1 AKI, 34 (31%) had stage 2 AKI, and 24 (22%) had stage 3 AKI. Severe AKI (KDIGO stage 2 and 3) was developed in 58 patients (2.7%; 95%CI 2.1-3.5) and 78 patients (71%) reached their maximum KDIGO stage on the first day of admission to PICU.”

4. Table 1 shows that 90 patients (81.8% were classified as pre-renal AKI 20 (18.2%) as renal AKI. 

Commentary: based on the list of primary diseases this information is probably inaccurate.

R/ Done. We are in agreement with the comment of the reviewer. To determine the etiology of acute kidney injury, we need a detailed medical history, complete physical examination and diagnostic tests. 

In our study, the most frequent etiologies were 1) infectious (which probably causes hypovolemic or renal vasoconstriction), 2) depletion of intravascular volume (due to gastrointestinal losses such as diarrhea and vomiting) and 3) redistribution to the interstitial space (secondary to malnutrition).

We clarify we are presenting a: “Potential physiopathological mechanism of AKI”. Considering this limitation, we added a full paragraph in discussion about.

Quote: “Our study has several limitations to consider when interpreting our results. First, we differentiated intrinsic renal damage and pre-renal injury according to some physiological and clinical characteristics of the underlying diseases. The most frequent etiologies were 1) infectious (which probably causes hypovolemic or renal vasoconstriction); 2) depletion of intravascular volume (due to gastrointestinal losses, such as diarrhea and vomiting); and 3) redistribution to the interstitial space (probably secondary to malnutrition). We know this strategy may potentially introduce information bias in this classification. However, it is part of our usual clinical care.”

5. In the methodology section: Severe acute kidney injury corresponded to patients classified as KDIGO 2 and 3. The results session states that “Severe AKI developed in 58 patients (2.7%; 95%CI 2.1 to 3.5). information that is also on table 2 but the results description continues as follows “ The severity classification according to KDIGO was 25 patients (22.7%) in stage 1, 40 patients (36.4%) in stage 2 and 45 patients (40.9%) in stage 3, severe AKI affected in fact 40+45= 85 patients. Commentary: the authors should review their definitions and review the tables as well as the rest of the text.

R/ Corrected. We checked our definitions of AKI stages following the comments and suggestion of the reviewer. We adjusted all related details and checked all details in Tables and text. In addition, we adjusted and re-ordered our results section under the first paragraphs as follows:

 “According to the KDIGO definition, AKI was diagnosed in 110 patients with 5.2% (95% CI 4.3% - 6.2%) prevalence. Of the patients diagnosed, 52 (47%) had stage 1 AKI, 34 (31%) had stage 2 AKI, and 24 (22%) had stage 3 AKI. Severe AKI (KDIGO stage 2 and 3) was developed in 58 patients (2.7%; 95%CI 2.1-3.5) and 78 patients (71%) reached their maximum KDIGO stage on the first day of admission to PICU.”

We also, adjusted Table 2 to be more clear and avoid confusion of the readers.

6. The phrase “On average, the severe stage of AKI was higher in non-survivors than in survivors” could be written as “The mortality rate for patients with severe AKI was higher than in the mild /moderate cases. 

The other parameters such as length of stay, mechanical ventilation and RRT could have their relation to AKI re-phrased in the same way English language should be revised throughout the text.

R/ Done. We have adjusted our description of the results following the recommendations. Also, we checked details about other outcomes.

Quote: “The mortality rate for patients with severe AKI was higher than in mild/moderate cases.”.

Quote: “An increase in the severity stage of AKI showed an incremental length of PICU stay (P <0.001) (Fig. 1). Median length of PICU stay was nine days for patients with stage 1 AKI; 10 days for stage 2 AKI, and 12 days for stage 3 AKI.”.

Quote: “The mean time of mechanical ventilation was 4.7 days for patients with a stage 1 AKI, 5.7 days for stage 2, and 8.5 days for stage 3.”.

7. The authors conclusions are in consonance with the literature but patient characteristics differ from the ones previously described in developed countries, the patients described in this study were admitted to medium complexity PICUs, mostly with medical primary diseases, 71% of the patients had their maximum KDIGO on day one of admission, so the diagnosis of AKI was probably late (is it due to lack of hospital facilities in the Colombian rural area?) and probably related to disorders of volume status and infection, almost 40% of them were presented malnutrition or were at risk of this comorbidity. 

My suggestion would be for the authors to describe these characteristics in more detail and the efforts that should be developed to minimize them.

R/ Done. Thank you for the comment. Indeed, these are major hospital which receive population from urban and many rural areas around (45% of the total). A common characteristic of those rural patients is the long-distances (and long-time) to reach appropriate medical care. Certainly, this may influence the state of AKI at admission but also the underlying mechanism of pre-renal vs renal AKI.

Following your comment, we added a complete paragraph to our discussion regarding our population details. 

Quote: “Pediatric patients with AKI reach their maximum KDIGO stage during the first week after diagnosis [9,16,25,26]. In our study, most of the patients included reached their maximum KDIGO stage on day one after admission to PICU and this may be explained by several potential factors: patients included came from rural areas, had varying degrees of malnutrition, and presented insurance-related barriers to health care in agreement with previous recent reports from Latin America [21].

The area of influence of most hospitals included is the department of Cauca and this is one of the least developed and rural areas in the country and among the most affected by the armed conflict [27]. It comprises 84 indigenous areas of the 679 existing in Colombia and 23% of its population has some alteration in the nutritional status [28]. In addition, the department of Cauca has only three hospitals of moderate complexity, which generates difficulty of access to health that include long distances and long time to reach appropriate care. Considering these characteristics, our study suggests that AKI diagnosis in our population was probably late and perhaps related to disorders of volume status, infection, degrees of malnutrition, and characteristics of the Colombian health care system. Infectious diseases are the most common related causes of AKI in developing countries in children [29]. In agreement, our results and other developing countries report similar profiles upon admission [24,25].”.

Reviewer #2: 

The authors conducted a prospective, observational study to assess the incidence and prognosis of AKI in PICUs of southeast Colombia. They found AKI of 5.2% and that AKI was associated with worse clinical outcomes.

The study will improve the knowledge of the local epidemiology of AKI in Colombia, but will not have any impact on the knowledge of what is already know about AKI and outcomes in pediatric population.

R/ Thanks for the comment. 

Our research allows us to know the regional epidemiology of AKI in pediatric patients. Probably we are not adding new information about AKI worldwide but our research allows us to describe the regional epidemiology and know the prognosis of AKI in pediatric Colombian patients and contribute to epidemiology in Latin America -where there are no many related studies-. 

Results of this research, mortality and associated factors can be –in some extent- extrapolated in medium and low complexity hospitals in low and middle income countries with characteristics similar to Colombia. In Latin America the epidemiology of AKI is poorly represented but it is under study. Also, it is a major reason to support the 0by25 initiative as we describe in our discussion.

1. I would recommend avoiding abbreviations on the abstract, and if used please define them (i.e. KDIGO).

R/ Done. 

We explain the abbreviation KDIGO. It is cited several times in our result section of our abstract and we consider it is appropriate.

2. Authors need to exclude from their analysis patients would were diagnosed with AKI at ER, and only include patient that developed AKI during PICU stay, the reasoning for this is that some % of patients with AKI diagnosis at ER could have non-diagnosed CKD, could be with AKD, or have community acquired AKI and this different type of populations have different risk for several short and long-term outcomes.

R/ Done. We clarify our recruitment process in our methods section. Patients were screened at Emergency department by a pediatric intensivist and very soon after diagnosis of AKI all patients were transferred for their treatment to PICU. This time-period is usually inferior to hours. In that sense, all our diagnosed patients were treated inside of the four PICU.

We know some patients could potentially be diagnosed early in time but due to characteristics of our population and the seeking of care process. The four included hospitals do not attend complex patients (i.e. cardiac surgery or transplant patients).

Considering the comment, we adjusted our objective mainly to describe this population with AKI and their outcomes. We describe only the frequency of AKI at admission to PICU (as a prevalence estimate) due to the methodological limitations.

3. Why do authors exclude patients with CKD? Since CKD is an important risk factor for subsequent AKI and could affect outcomes?

R/ Done.

Our aim was to evaluate patients of low and medium risk because of our population profile. In a previous comment, we mentioned that these four hospital are medium complexity and chronic patients are not usually admitted and send to a higher level of care. Considering this specific situation, we excluded patients with CKD.

4. How you deal with the fact that some patients did not had baseline serum creatinine? You used urine volume criteria? You used MDRD <60 ml/m/1.73m2 to estimate baseline sCr?

R/ Done and clarified. 

Our patients were admitted to PICU soon after AKI diagnosis. In order to screen and diagnose, we used baseline creatinine or the available measurement from emergency dept. We clarified our section improving details about the data collection.

Quote: “Baseline creatinine was the lowest value existing during six months prior to admission to PICU or that available at the emergency department.”

5. On table one; how you were able to differentiate patients with intrinsic renal damage from those who have Pre-renal? Please clarify. What do you refer that 18% of patients have “renal classification of AKI” this is not clear.

R/ Corrected. 

As mentioned in a previous comment, the renal and prerenal categorization was performed under clinical parameters. Those parameters, may introduce potential bias into this classification. In absence of other information (not widely used in clinical practice in Colombia) like urinary sodium, urinary osmolality, fractional excretion of sodium (EFNa), others. 

We clarify this potential limitation in our discussion section and to clarify our Table 1, we added a special subtitle: “Potential physiopathological mechanism of AKI”.

Quote “Our study has several limitations to consider when interpreting our results. First, we differentiated intrinsic renal damage and pre-renal injury according to some physiological and clinical characteristics of the underlying diseases. The most frequent etiologies were 1) infectious (which probably causes hypovolemic or renal vasoconstriction); 2) depletion of intravascular volume (due to gastrointestinal losses, such as diarrhea and vomiting); and 3) redistribution to the interstitial space (probably secondary to malnutrition). We know this strategy may potentially introduce information bias in this classification. However, it is part of our usual clinical care.”.

6. On page 8 you said that 58 patients developed severe AKI, however you later said that 45 patients developed stage 3 of AKI is that not consider severe AKI too? You need to check your numbers.

R/ Done. Corrected. We adjusted in detail classification of AKI Stages following your comment. Also, we checked all out text. This comment was made also for the reviewer #1 and we tried to solve that in detail.

Quote: “According to the KDIGO definition, AKI was diagnosed in 110 patients with 5.2% (95% CI 4.3% - 6.2%) prevalence. Of the patients diagnosed, 52 (47%) had stage 1 AKI, 34 (31%) had stage 2 AKI, and 24 (22%) had stage 3 AKI. Severe AKI (KDIGO stage 2 and 3) was developed in 58 patients (2.7%; 95%CI 2.1-3.5) and 78 patients (71%) reached their maximum KDIGO stage on the first day of admission to PICU.”.

7. You need to show fluid balance of your patients, fluid overload is an important contributing factor for adverse outcomes: Fluid overload should be assessed as % increase of body weight, Bouchard et al have shown that patients with fluid overload defined as an increase in body weight of over 10% had significantly more respiratory failure, need of mechanical ventilation, and more sepsis.

Please review and use this definition for fluid overload: Percentage of fluid overload adjusted for body weight: cumulative fluid balance that is expressed as a percent. A cutoff of ≥10% has been associated with increased mortality. Fluid overload percentage can be calculated using the following formula:

% Fluid overload = ((total fluid in – total fluid out)/admission body weight x 100)

Authors should stratified each group of patients in < 10% of fluid overload, 11% to 20% of fluid overload, > 20% of fluid overload for example, in order to assess for adverse outcomes. Review this study: Bouchard J, Mehta RL. Fluid balance issues in the critically ill patient. Contrib Nephrol. 2010;164:69-78.

What was the mean or median of fluid received during PICU stay? Would like to see fluids that were given by oral route and by I.V.

R/ Done. We know the importance of the association of fluid overload and increased morbidity and mortality in acute renal failure. However, as we discussed in the limitations, we did not collect data related in our study. Unfortunately, some data is missing in our daily clinical practice and it was part of our improvement process. In addition, other variables (i.e. renal ultrasound in patients with AKI) were also not widely reported and we describe these as limitations in our discussion section. 

Quote: “Secondly, we were unable to measure the percentage of fluid overload, despite the importance of the association of fluid overload and increased morbidity and mortality in acute renal failure [34]. We lack of a systematic record of these data in the clinical setting, along with other information, i.e., renal ultrasound in all patients with AKI diagnosis. Considering that, we introduced improved measures of clinical care. Third, difficulties emerged in the follow up of patients once they were discharged from the institution. We lost the possibility of contacting them for periodic assessment by pediatric nephrology and/or pediatrics and detect long-term outcomes, like the appearance of arterial hypertension or proteinuria, described widely as important markers of kidney disease. Our efforts to continue studying this population are undergoing. Finally, incidence of few events in some outcomes prevent further analysis and potential prognostic information remains under study.”.

9. I recommend also that authors need to look at factors that could have influence on having prolonged mechanical ventilation like sepsis, comorbidities (i.e. chronic illness), nutrition and metabolic problems, cardiovascular disease, pneumonia, and ventilation modality.

R/ Thanks for the comment. 

Our aim was to estimate and describe the prognosis of AKI diagnosed patients under our regional and local settings. There are no many multicenter studies conducted in middle-income countries related to AKI prognosis. We described some outcomes of patients diagnosed with AKI i.e. mortality, renal replacement therapy, use of drugs, use of mechanical ventilation, length of stay. 

Considering our recruited cohort and the low number of patients under some outcomes (i.e. mortality) we did not provide any other estimate about influencing factors –including PRISM score-. We did not find strong differences in use of mechanical ventilation (or days of mechanical ventilation) among severe vs mild/moderate cases of AKI and we avoid to analyze further those outcomes. Some international studies provide prior data regarding those outcomes and we are presenting some details in our discussion.

We hope to continue our efforts of active recruitment of this cohort and probably in the future we will provide strong data regarding these outcomes with more precise estimates. 

Thank you so much for the review process.

---

## [Decision Letter · Decision Letter 1]

2 Jun 2020

PONE-D-20-03806R1

Acute Renal Failure in Children. Multicenter Prospective Cohort Study in Medium Complexity Intensive Care Units from the Colombian Southeast

PLOS ONE

Dear Dr. FORERO,

Thank you for submitting your manuscript to PLOS ONE. After careful consideration, we feel that it has merit but does not fully meet PLOS ONE’s publication criteria as it currently stands. Therefore, we invite you to submit a revised version of the manuscript that addresses the points raised during the review process.

We look forward to receiving your revised manuscript.

Kind regards,

Emmanuel A Burdmann

Academic Editor

PLOS ONE

Reviewers' comments:

Reviewer's Responses to Questions

**Comments to the Author**

1. If the authors have adequately addressed your comments raised in a previous round of review and you feel that this manuscript is now acceptable for publication, you may indicate that here to bypass the “Comments to the Author” section, enter your conflict of interest statement in the “Confidential to Editor” section, and submit your "Accept" recommendation.

Reviewer #1: All comments have been addressed

Reviewer #2: All comments have been addressed

2. Is the manuscript technically sound, and do the data support the conclusions?

Reviewer #1: Yes

Reviewer #2: Yes

3. Has the statistical analysis been performed appropriately and rigorously? 

Reviewer #1: Yes

Reviewer #2: Yes

4. Have the authors made all data underlying the findings in their manuscript fully available?

Reviewer #1: Yes

Reviewer #2: Yes

5. Is the manuscript presented in an intelligible fashion and written in standard English?

Reviewer #1: Yes

Reviewer #2: Yes

6. Review Comments to the Author

Reviewer #1: The revised manuscript shows improvements in relation to the previous one

Title, abstract and introduction are clear

Methodology: the term creatinine serum has to be substituted by serum creatinine

Results: where it says " No significant association was noted for gender, age, primary diagnosis, and nutritional

status with the presence of severe AKI (Table 2" it should say No significant association was noted for gender, age, primary diagnosis, and nutritional

status with the presence of severe AKI (Table 1)

Table 2 is confusing as it classifies the patients into mild/moderate and severe AKI and all the p values are non significant , but the text follows with a paragraph where it is clear that the significant differences were mainly between KDIGO 1 and 3 patients, so table 2 has to be modified and present data classified according to KDIGO status 1, 2 and 3, incorporating all the significant p values shown in the following paragraphs

Discussion is much improved and shows the importance of the presented data to Latin America

The text still needs an English language revision

Reviewer #2: (No Response)

7. PLOS authors have the option to publish the peer review history of their article (what does this mean?). If published, this will include your full peer review and any attached files.

Reviewer #1: Yes: Vera Koch

Reviewer #2: No

---

## [Author Response · Author response to Decision Letter 1]

21 Jun 2020

Professor

Emmanuel A Burdmann

Academic Editor

PLOS ONE

I would like to acknowledge for the special and learning-based reviews received in our manuscript (PONE-D-20-03806R1) entitled “Acute Renal Failure in Children. Multicenter Prospective Cohort Study in Medium Complexity Intensive Care Units from the Colombian Southeast”. 

We tried to solve all comments of the second review and follow all suggestions of the reviewers to improve the quality and transparency of our paper in this second round. We carefully considered all reviewers’ comments and we are submitting a detailed list of responses to each comment with quotes from the text. 

We hope that our manuscript can fully meet all the PLOS ONE’s publication criteria.

Sincerely,

Jessica M. Forero on behalf of the authors

Detailed responses to each comment.

1. If the authors have adequately addressed your comments raised in a previous round of review and you feel that this manuscript is now acceptable for publication, you may indicate that here to bypass the “Comments to the Author” section, enter your conflict of interest statement in the “Confidential to Editor” section, and submit your "Accept" recommendation.

Reviewer #1: All comments have been addressed

Reviewer #2: All comments have been addressed

Thank you for the comment.

2. Is the manuscript technically sound, and do the data support the conclusions?

The manuscript must describe a technically sound piece of scientific research with data that supports the conclusions. Experiments must have been conducted rigorously, with appropriate controls, replication, and sample sizes. The conclusions must be drawn appropriately based on the data presented

Reviewer #1: Yes

Reviewer #2: Yes

Thank you for the comment.

3. Has the statistical analysis been performed appropriately and rigorously?

Reviewer #1: Yes

Reviewer #2: Yes

Thank you for the comment.

4. Have the authors made all data underlying the findings in their manuscript fully available?

The PLOS Data policy requires authors to make all data underlying the findings described in their manuscript fully available without restriction, with rare exception (please refer to the Data Availability Statement in the manuscript PDF file).

Reviewer #1: Yes

Reviewer #2: Yes

Thank you for the comment.

5. Is the manuscript presented in an intelligible fashion and written in standard English?

Reviewer #1: Yes

Reviewer #2: Yes

Thank you for the comment.

Reviewer #1

The revised manuscript shows improvements in relation to the previous one: Title, abstract and introduction are clear

Done. Thank you for the comment.

Methodology: the term creatinine serum has to be substituted by serum creatinine

Done. The term was replaced in three situations in the methodology section.

Results: where it says " No significant association was noted for gender, age, primary diagnosis, and nutritional status with the presence of severe AKI (Table 2" it should say No significant association was noted for gender, age, primary diagnosis, and nutritional status with the presence of severe AKI (Table 1)

Done. Corrected.

Table 2 is confusing as it classifies the patients into mild/moderate and severe AKI and all the p values are non-significant, but the text follows with a paragraph where it is clear that the significant differences were mainly between KDIGO 1 and 3 patients, so table 2 has to be modified and present data classified according to KDIGO status 1, 2 and 3, incorporating all the significant p values shown in the following paragraphs.

Done. As we stated in our methods section Table 1 presents results for mild/moderate KDIGO (st 1) versus severe AKI (st 2-3) measured at admission. Following the reviewer's recommendation, we added two columns to present independently KDIGO stages 1, 2, and 3 and adjusted the next paragraph of the result section called “Association between severity of Acute Kidney Injury on admission to PICU and mortality”. 

However, the following paragraphs present details regarding outcomes about the “maximum KDIGO stage reached during the PICU stay”, and not only at admission. To clarify this difference, we adjusted the titles very clear to each section and also we added the Word “maximum KDIGO stage” to each result (highlighted).

• Table 2 title: Outcomes in patients with AKI diagnosis stratified by KDIGO stage, mild/moderate AKI and severe AKI on admission.

• Paragraph 1 subtitle: Association between severity of Acute Kidney Injury on admission to PICU and mortality.

• Paragraph 2 subtitle: Maximum Stage of Acute Kidney Injury during PICU stay and Outcomes.

Discussion is much improved and shows the importance of the presented data to Latin America

The text still needs an English language revision.

Done. Thank you for the comment.

Finally, the manuscript was revised again in detail to enhance their language style and grammar.

---

## [Decision Letter · Decision Letter 2]

26 Jun 2020

Acute Renal Failure in Children. Multicenter Prospective Cohort Study in Medium Complexity Intensive Care Units from the Colombian Southeast

PONE-D-20-03806R2

Dear Dr. Forero,

We’re pleased to inform you that your manuscript has been judged scientifically suitable for publication and will be formally accepted for publication once it meets all outstanding technical requirements.

Kind regards,

Emmanuel A Burdmann

Section Editor

PLOS ONE

Additional Editor Comments (optional):

Reviewers' comments:

Reviewer's Responses to Questions

**Comments to the Author**

1. If the authors have adequately addressed your comments raised in a previous round of review and you feel that this manuscript is now acceptable for publication, you may indicate that here to bypass the “Comments to the Author” section, enter your conflict of interest statement in the “Confidential to Editor” section, and submit your "Accept" recommendation.

Reviewer #1: All comments have been addressed

2. Is the manuscript technically sound, and do the data support the conclusions?

Reviewer #1: (No Response)

3. Has the statistical analysis been performed appropriately and rigorously? 

Reviewer #1: (No Response)

4. Have the authors made all data underlying the findings in their manuscript fully available?

Reviewer #1: (No Response)

5. Is the manuscript presented in an intelligible fashion and written in standard English?

Reviewer #1: (No Response)

6. Review Comments to the Author

Reviewer #1: (No Response)

7. PLOS authors have the option to publish the peer review history of their article (what does this mean?). If published, this will include your full peer review and any attached files.

Reviewer #1: **Yes: **Vera Koch

---

## [Editor Report · Acceptance letter]

12 Aug 2020

PONE-D-20-03806R2 

Acute Renal Failure in Children. Multicenter Prospective Cohort Study in Medium-Complexity Intensive Care Units from the Colombian Southeast 

Dear Dr. Forero-Delgadillo:

I'm pleased to inform you that your manuscript has been deemed suitable for publication in PLOS ONE. Congratulations! Your manuscript is now with our production department. 

Kind regards, 

on behalf of

Dr. Emmanuel A Burdmann 

Section Editor

PLOS ONE